# IoT and Ensemble Long-Short-Term-Memory-Based Evapotranspiration Forecasting for Riyadh

**DOI:** 10.3390/s23177583

**Published:** 2023-09-01

**Authors:** Muhammad Asif Nauman, Mahlaqa Saeed, Oumaima Saidani, Tayyaba Javed, Latifah Almuqren, Rab Nawaz Bashir, Rashid Jahangir

**Affiliations:** 1Department of Computer Science, University of Engineering and Technology, Lahore 54890, Pakistan; asif.nauman.uet@gmail.com; 2Department of Computer Science, University of South Asia, Lahore 53400, Pakistan; mahlaqa.saeed@usa.edu.pk; 3Department of Information Systems, College of Computer and Information Sciences, Princess Nourah bint Abdulrahman University, P.O. Box 84428, Riyadh 11671, Saudi Arabia; laAlmuqren@pnu.edu.sa; 4Department of Computer Science, Barani Institute of Information Technology, Rawalpindi 46604, Pakistan; tayyaba@biit.edu.pk; 5Department of Computer Science, COMSATS University Islamabad, Vehari Campus, Vehari 61100, Pakistan; rabnawaz@cuivehari.edu.pk (R.N.B.); rashidjahangir@cuivehari.edu.pk (R.J.)

**Keywords:** ensemble learning, long short-term memory (LSTM), evapotranspiration (*ET*), boosted LSTM, bagged LSTM

## Abstract

Evapotranspiration (ET) is the fundamental component of efficient water resource management. Accurate forecasting of ET is essential for efficient water utilization in agriculture. ET forecasting is a complex process due to the requirements of large meteorological variables. The recommended approach is based on the Internet of Things (IoT) and an ensemble-learning-based approach for meteorological data collection and ET forecasting with limited meteorological conditions. IoT is part of the recommended approach to collect real-time data on meteorological variables. The daily maximum temperature (T), mean humidity (Hm), and maximum wind speed (Ws) are used to forecast evapotranspiration (ET). Long short-term memory (LSTM) and ensemble LSTM with bagged and boosted approaches are implemented and evaluated for their accuracy in forecasting ET values using meteorological data from 2001 to 2023. The results demonstrate that the bagged LSTM approach accurately forecasts ET with limited meteorological conditions in Riyadh, Saudi Arabia, with the coefficient of determination (R${}^{2}$) of 0.94 compared to the boosted LSTM and off-the-shelf LSTM with R${}^{2}$ of 0.91 and 0.77, respectively. The bagged LSTM model is also more efficient with small values of root mean squared error (RMSE) and mean squared error (MSE) of 0.42 and 0.53 compared to the boosted LSTM and off-the-shelf LSTM models.

## 1. Introduction

Evapotranspiration (*ET*) is a significant process of the water cycle [1]. The objective of smart agriculture is to optimize resource management and maximize crop yields with efficient use of resources, especially irrigation water [2]. The smart irrigation system is the basis for achieving high yields from crops with scarce irrigation water resources. The implementation of irrigation water according to *ET* is the foundation of efficient irrigation water. The global scarcity of freshwater resources emphasizes the urgent need for efficient utilization of water resources. The efficiency of the irrigation system is highly dependent on the *ET* process. The accurate forecasting of *ET* can result in better agricultural management through water resource planning and drought management [3,4].

*ET* is the transformation of water into vapor, called evaporation, and the release of water vapor from the surface of plants, called transpiration [5]. *ET* is crucial in maintaining the water balance between soil and atmosphere [6]. *ET* helps to balance water levels on the Earth’s surface and underground [7]. *ET* maintains the overall hydrological balance in ecosystems. Meteorological conditions like daily maximum temperature (*T*), mean humidity (*Hm*), carbon dioxide ($CO_{2}$) concentration, air moisture level, and maximum wind speed (*Ws*) significantly affect the *ET* [8,9,10]. The major meteorological factors affecting *ET* are illustrated in Figure 1.

The Penman–Montieth (PM) method is an accurate method of *ET* calculation and is also approved by the Food and Agriculture Organization (FAO) of the United Nations (UN) for *ET* calculations [11]. The PM method is expressed by Equation (Equation 1) [12]. The PM method is a widely recognized method of *ET* calculations using the meteorological data of a location. PM calculates the evaporation and transpiration rate for specific meteorological conditions. The PM method is the most accurate method of *ET* calculation [13]. The major issue of the PM method is its requirement for the large number of meteorological variables that make it difficult to use in hydrological cycles [11].
(1)$$ET=\frac{0.408m\left(R-G\right)+c\frac{900}{Tm+273}Ws\left(es-ea\right)}{m+c(1+0.0.34Ws)}$$
where *ET* depicts the evapotranspiration in mm/day, m depicts the slope of the vapor pressure curve in kPa/°C, R depicts the net radiation at the crop surface in MJ/m²/day, G depicts the soil heat flux density in MJ/m²/day, c depicts the psychrometric constant in kPa/°C, Tm depicts the mean daily air temperature in °C, Ws depicts the wind speed at 2 m height in m/s, es depicts the saturated vapor pressure in kPa, and ea depicts the actual vapor pressure in kPa. The *ET* determination with a large number of meteorological inputs is a complex process [14]. The acquisition of large amounts of data are costly and time-consuming. Simplification of *ET* forecasting with limited meteorological data are important for the practical implementation of *ET*-based irrigation water applications.

The Internet of Things (IoT) is widely utilized in modern technology in many applications. IoT provides improved efficiency, automation, optimization, real-time data collection, and monitoring [1]. IoT also has made revolutionary changes in agriculture, precision farming, energy management, industrial automation, and environmental monitoring [15,16,17,18]. IoT can address many irrigation issues faced in traditional agricultural practices. IoT can be utilized in smart irrigation systems [19]. The ability of IoT to capture and integrate contextual factors can be utilized to provide better forecasting of *ET* rates [20]. IoT devices and sensors can collect real-time data on meteorological conditions to make different contextual decisions in agriculture [21]. IoT systems provide valuable inputs for accurate and precise *ET* forecasting models.

Machine learning (ML) with IoT has a wide range of applications in precision agriculture. ML has many advantages that can provide better suitability and efficiency integrated with IoT context-oriented applications [22]. ML algorithms can handle large volumes of IoT-generated data. ML can identify patterns and relationships between meteorological variables to make decisions [23,24]. ML can be utilized for the accurate forecasting of *ET* [20]. From this perspective, this paper proposes an IoT and ensemble learning-based approach for the forecasting of *ET* in Riyadh. Three meteorological variables, daily maximum temperature (*T*), mean humidity (*Hm*), and maximum wind speed (*Ws*), are considered for *ET* forecasting in the proposed approach. The data are collected through the proposed long-range-wireless-area-network (LoRaWAN)-enabled IoT architecture. Long-short-term-memory (LSTM)-based ensemble ML approaches are part of the proposed solution to assess the performance of ML models in handling the complex relationship between meteorological conditions and *ET*. The main contributions of the study are as follows:LoRaWAN-enabled IoT architecture to sense meteorological conditions is proposed and implemented to predict accurate *ET* according to real-time meteorological conditions.Off-the-shelf LSTM and bagged and boosted ensemble LSTM ML approaches are implemented and evaluated to forecast the *ET* values from real-time meteorological data collected through the proposed LoRaWAN-enabled IoT architecture.The evaluation of the performance of ensemble LSTM approach and off-the-shelf LSTM ML models for *ET* forecasting for Riyadh in Saudi Arabia.

IoT is effectively used in acquiring the parameters affecting *ET* values. Moreover, ML algorithms with IoT are proven to be very efficient in forecasting *ET* values [22,25]. The accurate forecasting of *ET* can lead to better irrigation and agriculture productivity [26]. IoT and ML-based approaches for *ET* forecasting are discussed in this section.

Hu et al. [20] proposed an ML model using temperature and humidity. They apply the Internet of Things (IoT) architecture for accurately estimating evapotranspiration (*ET*) for efficient irrigation water management and conservation. They used crop field data from Pakistan from 2015 to 2016. The results suggested that the k-nearest neighbors (KNN) model outperformed the other models and achieved 92% accuracy. Moreover, KNN is successful in reducing the root mean squared error (RMSE) and mean absolute error (MAE) by 16% and 3%, respectively.

Nawandar et al. [27] proposed an artificial-neural-network (ANN)-based model to forecast *ET* for maximum crop yield considering water loss in the irrigation system. The proposed approach is compared with the PM method with fewer inputs. The proposed model successfully produces comparable results to the PM method, with a maximum error of only 0.4 mm day${}^{-}$${}^{1}$ in the forecasting of *ET*.

Bellido-Jiménez et al. [28] proposed an ML-based approach for the accurate forecasting of *ET* for regions with limited water resources. The methodology considered two inputs, EnergyT and Hourmin. It was observed that multilayer perceptron (MLP) and extreme learning machine (ELM) performed better across all sites with a determination factor greater than 0.89.

Bashir et al. [14] proposed an ensemble ANN-based approach for estimating *ET* for water conservation in agriculture. They implemented three models named Model-1 (ensemble model), Model-2 (ANN model), and Model-3 (ANN model temperature input only). Model-1 outperformed other models and achieved an accuracy of 91.44%. The predictions from Model-1 also exhibit a stronger correlation with the *ET* values obtained through the PM approach. Moreover, it was observed that Model-1 has a Pearson correlation coefficient (r) of 0.996. However, Model-2 and Model-3 show weaker correlations.

Majumdar et al. [29] recommended an IoT and ML-oriented approach for the forecasting of *ET* at each stage of rice growth. For *ET* prediction, individual ML models and ensemble learning methods are compared. The results show that individual models suffer from high bias and variance, leading to inadequate prediction accuracy. From their research, it is evident that boosting is the most effective approach for *ET* and soil moisture prediction in different growth stages of rice cultivation. It outperforms other models in terms of accuracy, as demonstrated by lower error metrics and higher determination coefficients, to support better water management decisions throughout the crop cycle.

Torres et al. [30] proposed an IoT-based multilevel data fusion approach, Hydra. They predicted irrigation needs based on soil moisture levels and estimated crop *ET* to forecast the appropriate irrigation timing. Moreover, a novel *ET* model was generated using the quadratic support vector machine (SVM) based model. The results indicate that Hydra, with its multilevel data fusion approach, improves sensor accuracy, identifies target events more accurately, and enables better decision-making.

Chen Zhijun et al. [31] assessed the deep learning approaches named deep neural network (DNN), temporal convolution neural network (TCN), and LSTM to forecast daily *ET* using limited climate data in China. The performance of the deep learning approach was compared with the ML model and empirical equations. The results showed that TCN and RF outperformed daily *ET* measurements in China. Jia Weibing et al. [32] introduced two novel hybrid models, combining particle swarm optimization (PSO) with LSTM neural network to forecast *ET* at four stations in China. Jung Dae-Hyun et al. [33] recommended deep-learning approaches for *ET* prediction in tomato greenhouses. The LSTM model yields low RMSE values of 0.00356 in *ET* forecasting. Roy Dilip Kumar et al. [34] proposed the daily and multistep forward *ET* prediction model using LSTM and bi-directional LSTM (Bi-LSTM) models. The LSTM model outperformed the support vector regressor (SVR) for daily *ET* predictions. Yin Juan et al. [35] proposed a hybrid Bi-LSTM approach to predict daily *ET*. The proposed model is successfully integrated into an intelligent irrigation system in China’s central Ningxia region with limited meteorological data. Wang Tianteng et al. [36] proposed *ET* for fruit tree farms.

It is concluded that much research is conducted for the forecasting of *ET* in precision agriculture. The crop yield and productivity can be increased by the automation of *ET* monitoring and smart irrigation systems. In this perspective, IoT and ML-based approaches are proven to be very effective. This research proposes an ensemble learning-based model with bagged LSTM and boosted LSTM for *ET* forecasting in Riyadh.

The interactions between meteorological conditions and *ET* are diverse in nature and nonlinear. These relationships between meteorological conditions and *ET* also change from location to location. Therefore, to handle the complexity between meteorological conditions and *ET*, an LSTM-based ML approach is proposed. To deal with the dynamic nature of the problem, there is also a need to handle seasonal variations. Keeping in view the above issues in *ET* forecasting using the LSTM algorithm to handle the complex relationship between the *ET* and meteorological conditions. LSTM can handle seasonal variations and patterns better due to its ability to handle sequential data with memory capabilities. *ET* forecasting with limited meteorological conditions for Riyadh, Saudi Arabia, is proposed.

The subsequent sections of this research article are structured into materials and methods, results, discussion, and conclusion. The materials and methods section illustrates the flow chart, LoRaWAN-enabled IoT architecture, the data set used for the study, and ML implementation. The results and discussion sections evaluate the performance of the model in terms of different metrics. The conclusion section concludes the significant findings and implications of the proposed solution.

## 2. Materials and Methods

An IoT and ensemble LSTM-oriented ML-based approach for *ET* forecasting in Riyadh, Saudi Arabia, is proposed with limited meteorological conditions. The meteorological conditions in this regard are daily maximum temperature (*T*), mean humidity (*Hm*), and maximum wind speed (*Ws*). The proposed approach utilizes real-time data collected through the proposed LoRaWAN-enabled IoT architecture. The real-time meteorological data are used to train the off-the-shelf LSTM and ensemble LSTM models with bagging and boosting approaches. The flowchart of the IoT-ML-based proposed approach is presented in Figure 2.

The main steps of the flow chart for the forecasting of *ET* are as follows:First, meteorological data (daily maximum temperature (*T*), mean humidity (*Hm*), and maximum wind speed (*Ws*)) were determined from daily sensed temperature, humidity, and wind speed using LoRaWAN-enabled proposed IoT architecture.Second, the real-time data are preprocessed by cleaning and normalization to bring all variables to a similar scale.From the real-time collected data, the off-the-shelf LSTM and ensemble LSTM-based bagged and boosted ML models are trained.Finally, the performance of both ensemble LSTM models for *ET* forecasting in Riyadh is evaluated.

The simplification of the *ET* forecasting process is essential for its successful implementation in smart agriculture applications [20]. Simplification of *ET* forecasting with fewer meteorological conditions is proposed to deal with the difficulty associated with *ET* forecasting. The *ET* forecast based on daily maximum temperature (*T*), mean humidity (*Hm*), and maximum wind speed (*Ws*) is proposed for *ET* forecasting in Riyadh, Saudi Arabia. An IoT ML-based model is implemented to forecast *ET* using daily maximum temperature (*T*), mean humidity (*Hm*), and maximum wind speed (*Ws*) meteorological conditions, which are expressed by Equation (Equation 2).
(2)ET→f(T,Hm,Ws)
where *ET* is the evapotranspiration, T is the daily maximum temperature, Hm is the mean humidity, and Ws is the daily maximum wind speed. The proposed LoRaWAN-enabled IoT architecture for sensing meteorological conditions is shown in Figure 3. The sensing meteorological data from the sensor nodes are transmitted to the server through LoRaWAN. Real-time data on meteorological variables (daily maximum temperature (*T*), mean humidity (*Hm*), and daily maximum wind speed (*Ws*) were calculated from meteorological data collected from Riyadh, Saudi Arabia. For this purpose, IoT architecture with LoRaWAN is utilized. LoRaWAN technology is utilized for wireless communication between the sensors and the IoT gateway without the Internet. The architecture allows the collection of meteorological data from remote areas without involving the Internet. The data collected through the IoT sensors is transmitted to the central server through the LoRaWAN gateway. The meteorological data at the server is processed for the training of ML models. Apart from these functions, different services are also part of the server, like data storage, data analytics, data management, and server configuration, as shown in Figure 3.

### 2.1. Location

The proposed solution intends to forecast the *ET* of Riyadh, Saudi Arabia. Riyadh is situated in the central western region of Saudi Arabia at an altitude of 2000 feet from sea level with longitude and latitude of 24.7136${}^{\circ }$ N, 46.6753${}^{\circ }$ E, respectively, as shown in Figure 4. The meteorological conditions of Riyadh are desert in nature with semi-arid conditions. The desert climatic conditions pose challenges to the efficient use of irrigation water to support agricultural activities in such areas.

### 2.2. Data Set

The meteorological data were acquired in Riyadh from the year 2001 to 2023 to forecast the *ET* according to the PM approach. The meteorological data were used to determine the *ET* by PM approach to develop the data set. To simplify the *ET* determination process, this data set is utilized to train the ML model for *ET* forecasting using only daily maximum temperature (*T*), mean humidity (*Hm*), and daily maximum wind speed (*Ws*). The meteorological data selection for *ET* forecasting is made based on their strong correlation with *ET*, shown in Figure 5. The daily maximum temperature (*T*), mean humidity (*Hm*), and maximum wind speed (*Ws*) are strongly correlated with *ET*; therefore, these meteorological conditions are chosen.

The *ET* distribution in the data set calculated by the PM approach for each month is displayed in Figure 6. It is noted that the *ET* for each month exhibits a similar pattern over each year.

The relationship between *ET* and daily maximum temperature (*T*) for each month from the year 2001 to March of 2023 is shown in Figure 7. The daily maximum temperature (*T*) has a positive relationship with *ET*. The relationship between *ET* and daily mean humidity (*Hm*) for each month from the year 2001 to March of 2023 is shown in Figure 8. The daily mean humidity (*Hm*) has a negative relationship with *ET*. The relationship between *ET* and wind speed (Ws) for each month from the year 2001 to March of 2023 is shown in Figure 9. The wind speed (Ws) has a positive relationship with *ET*. To deal with the non-linear nature of the problem, an ML-assisted solution is proposed to forecast *ET* with limited meteorological conditions.

### 2.3. Pre-Processing of Data

The collected data are in raw form and have much unnecessary information. The collected data were preprocessed before using them for ML models. The main steps for preprocessing are dealing with missing values, the imputation method, normalization, and standardization of data. The flow chart of the preprocessing is presented in Figure 10. The preprocessing of data can lead to reliable and efficient forecasting of *ET*. The data preprocessing is performed in the following manner.

First, the process of handling missing values is performed. The main reasons for these issues are sensor malfunctions or other data collection errors. Such missing values can cause bias in the model’s performance. Mean imputation is applied for the handling of missing values. In this method, the missing values are substituted with the average of the values.Second, the outliers are identified and handled from the raw data. Outliers are data values that have a significant deviation from the normal distribution of the variables. In the proposed approach, the sensed data are detected in their respective ranges. Data received out of these ranges is discarded.Third, the normalization process is implemented to make all variables of the data set on a similar scale. It is essential for the performance of the proposed model. The variation in different scale variables can cause inaccurate impacts on the model’s performance.Finally, the standardization technique is utilized for the normalization of the data that transforms the variables to have zero mean and unit variance.

### 2.4. Implementation of ML Models

In the proposed approach, the ensemble learning LSTM technique with off-the-shelf LSTM is utilized for accurate *ET* forecasting. The basic architecture of LSTM is presented in  Figure 11. For the implementation of the ML model, the Sciket-learn, seaborne, and Matplotlib libraries of Python are used. The data set is divided into training and testing sets using a 70:30 ratio.

The ensemble model tends to enhance the predictive accuracy in complex problems and reduce bias in model training [2,29]. The purpose of the implementation of the ensemble approach is to explore the possibilities of accurate *ET* forecasting with different configurations of the LSTM models.

#### 2.4.1. Bagged LSTM

Bagged LSTM employs an ensemble learning strategy where several LSTM models are trained using subsets of the training data. The bagged LSTM reduces over-fitting and enhances the model’s capacity to generalize. The algorithm of bagged LSTM is presented in Algorithm 1, Where, $h_{i}$, $c_{i}$, and *i* are hidden states, cell states, and time steps, respectively. The output equation calculates the predicted value $y_{pred_{i}}$ based on the hidden state $h_{i}$. The loss function $L_{i}$ assesses the disparity between the predicted value and the target $y_{i}$. The model parameters $W_{f},W_{i},W_{o},W_{c},W_{y},b_{f},b_{i},b_{o},b_{c},b_{y}$ are used to minimize the loss with the help of back-propagation and gradient descent. The implementation of the bagged approach is shown in  Figure 12 and by Algorithm 1.
**Algorithm 1** Bagged LSTM algorithm for *ET* forecasting **Require:**   Temperature sequence $T=[T_{1},T_{2},\ldots ,T_{n}]$   Humidity sequence $Hm=[Hm_{1},Hm_{2},\ldots ,Hm_{n}]$   Wind speed sequence $Ws=[Ws_{1},Ws_{2},\ldots ,Ws_{n}]$   Output sequence $ET=[ET_{1},ET_{2},\ldots ,ET_{n}]$   Number of LSTM models *M*   Number of epochs *E***Ensure:**   Bagged LSTM model weights $W=[w_{1},w_{2},\ldots ,w_{M}]$ 1:Initialize empty list *W* 2:**for** *m* in range(*M*) **do** 3:    Initialize LSTM model $LSTM_{m}$ 4:    Randomly split the data into training and validation sets 5:    Initialize input sequence $X=[\left(T_{1},Hm_{1},Ws_{1}\right),\left(T_{2},Hm_{2},Ws_{2}\right),\ldots ,\left(T_{n},Hm_{n},Ws_{n}\right)]$ 6:    Initialize output sequence $Y=[ET_{1},ET_{2},\ldots ,ET_{n}]$ 7:    Train $LSTM_{m}$ on the training set for *E* epochs using the following equations: 8:    **for** each epoch *e* in range(*E*) **do** 9:        **for** *i* in range(*n*) **do** 10:           Set the LSTM input sequence $x_{i}=\left(Ws_{i},Hm_{i},Ws_{i}\right)$ 11:           Set the LSTM target output $y_{i}=ET_{i}$ 12:           Calculate the LSTM hidden state $h_{i}$ and cell state $c_{i}$ using the LSTM equations: 13:           $f_{i}=\sigma \left(W_{f}\cdot \left[h_{i-1},x_{i}\right]+b_{f}\right)$ 14:           $i_{i}=\sigma \left(W_{i}\cdot \left[h_{i-1},x_{i}\right]+b_{i}\right)$ 15:           $o_{i}=\sigma \left(W_{o}\cdot \left[h_{i-1},x_{i}\right]+b_{o}\right)$ 16:           ${\tilde{c}}_{i}=\tanh\left(W_{c}\cdot \left[h_{i-1},x_{i}\right]+b_{c}\right)$ 17:           $c_{i}=f_{i}\cdot c_{i-1}+i_{i}\cdot {\tilde{c}}_{i}$ 18:           $h_{i}=o_{i}\cdot \tanh\left(c_{i}\right)$ 19:           Calculate the LSTM output $y_{pred_{i}}$ using the output equation: 20:           $y_{pred_{i}}=W_{y}\cdot h_{i}+b_{y}$ 21:           Calculate the LSTM loss $L_{i}$ using a suitable loss function (e.g., mean squared error): 22:           $L_{i}=\frac{1}{2}{\left(y_{pred_{i}}-y_{i}\right)}^{2}$ 23:           Update the LSTM model parameters $W_{f},W_{i},W_{o},W_{c},W_{y},b_{f},b_{i},b_{o},b_{c},b_{y}$ using back-propagation and gradient descent 24:    Evaluate the performance of $LSTM_{m}$ on the validation set 25:    Store the weights of $LSTM_{m}$ in the list *W* 26:Normalize the weights in *W* 27:**return** *W*

#### 2.4.2. Boosted LSTM

Boosted LSTM is an efficient ensemble learning approach where multiple LSTM models are trained sequentially. In this model, each subsequent LSTM model focuses on the samples that were unclassified by the previous models. This approach leads to improving the overall accuracy. Boosted LSTM combines the forecasting of the individual models using weighted averaging to obtain the final forecasting. The algorithm of boosted LSTM is presented in Algorithm 2 and in the flow chart shown in Figure 13.
**Algorithm 2** Boosted LSTM algorithm for ET forecasting. **Require:**   Temperature sequence $T=[T_{1},T_{2},\ldots ,T_{n}]$   Humidity sequence $Hm=[Hm_{1},Hm_{2},\ldots ,Hm_{n}]$   Wind speed sequence $Ws=[Ws_{1},Ws_{2},\ldots ,Ws_{n}]$   Output sequence $ET=[ET_{1},ET_{2},\ldots ,ET_{n}]$   Number of LSTM models *M*   Number of epochs *E***Ensure:**   Boosted LSTM model weights $W=[w_{1},w_{2},\ldots ,w_{M}]$ 1:Initialize empty list *W* 2:**for** *m* in range(*M*) **do** 3:    Initialize LSTM model $LSTM_{m}$ 4:    Randomly split the data into training and validation sets 5:    Initialize input sequence $X=[\left(T_{1},Hm_{1},Ws_{1}\right),\left(T_{2},Hm_{2},Ws_{2}\right),\ldots ,\left(T_{n},Hm_{n},Ws_{n}\right)]$ 6:    Initialize output sequence $Y=[ET_{1},ET_{2},\ldots ,ET_{n}]$ 7:    Train $LSTM_{m}$ on the training set for *E* epochs using the following equations: 8:    **for** each epoch *e* in range(*E*) **do** 9:        **for** *i* in range(*n*) **do** 10:           Set the LSTM input sequence $x_{i}=\left(T_{i},Hm_{i},Ws_{i}\right)$ 11:           Set the LSTM target output $y_{i}=ET_{i}$ 12:           Calculate the LSTM hidden state $h_{i}$ and cell state $c_{i}$ using the LSTM equations: 13:           $f_{i}=\sigma \left(W_{f}\cdot \left[h_{i-1},x_{i}\right]+b_{f}\right)$ 14:           $i_{i}=\sigma \left(W_{i}\cdot \left[h_{i-1},x_{i}\right]+b_{i}\right)$ 15:           $o_{i}=\sigma \left(W_{o}\cdot \left[h_{i-1},x_{i}\right]+b_{o}\right)$ 16:           ${\tilde{c}}_{i}=\tanh\left(W_{c}\cdot \left[h_{i-1},x_{i}\right]+b_{c}\right)$ 17:           $c_{i}=f_{i}\cdot c_{i-1}+i_{i}\cdot {\tilde{c}}_{i}$ 18:           $h_{i}=o_{i}\cdot \tanh\left(c_{i}\right)$ 19:           Calculate the LSTM output $y_{pred_{i}}$ using the output equation: 20:           $y_{pred_{i}}=W_{y}\cdot h_{i}+b_{y}$ 21:           Calculate the LSTM loss $L_{i}$ using a suitable loss function (e.g., mean squared error): 22:           $L_{i}=\frac{1}{2}{\left(y_{pred_{i}}-y_{i}\right)}^{2}$ 23:           Update the LSTM model parameters $W_{f},W_{i},W_{o},W_{c},W_{y},b_{f},b_{i},b_{o},b_{c},b_{y}$ using back-propagation and gradient descent: 24:           $W_{f}=W_{f}-\alpha \cdot \frac{\partial L_{i}}{\partial W_{f}}$ 25:           $W_{i}=W_{i}-\alpha \cdot \frac{\partial L_{i}}{\partial W_{i}}$ 26:           $W_{o}=W_{o}-\alpha \cdot \frac{\partial L_{i}}{\partial W_{o}}$ 27:           $W_{c}=W_{c}-\alpha \cdot \frac{\partial L_{i}}{\partial W_{c}}$ 28:           $W_{y}=W_{y}-\alpha \cdot \frac{\partial L_{i}}{\partial W_{y}}$ 29:           $b_{f}=b_{f}-\alpha \cdot \frac{\partial L_{i}}{\partial b_{f}}$ 30:           $b_{i}=b_{i}-\alpha \cdot \frac{\partial L_{i}}{\partial b_{i}}$ 31:           $b_{o}=b_{o}-\alpha \cdot \frac{\partial L_{i}}{\partial b_{o}}$ 32:           $b_{c}=b_{c}-\alpha \cdot \frac{\partial L_{i}}{\partial b_{c}}$ 33:           $b_{y}=b_{y}-\alpha \cdot \frac{\partial L_{i}}{\partial b_{y}}$ 34:    Evaluate the performance of $LSTM_{m}$ on the validation set 35:    Calculate the weight $w_{m}$ based on the validation performance of $LSTM_{m}$ 36:    $w_{m}=\frac{\mathrm{Validation}\;\mathrm{Accuracy}\;\mathrm{of}\;LSTM_{m}}{\mathrm{Total}\;\mathrm{Validation}\;\mathrm{Accuracy}}$ 37:    Store the weight $w_{m}$ in the list *W* 38:Normalize the weights in *W* 39:**return** *W*

## 3. Results

The performance of ML models for *ET* forecasting is assessed using the 30% data set with the following evaluation metrics:Coefficient of determination (R${}^{2}$);Pearson correlation coefficient (r);Root mean squared error (RMSE);Mean squared error (MSE).

R${}^{2}$ is the evaluation metric to judge the accuracy of *ET* forecasting with the ML model. R${}^{2}$ is expressed by Equation (Equation 3), where E${}_{i}$ is the actual *ET* value in the data set at the *i*-th data instance, $\hat{E}$${}_{i}$ is the forecasted *ET* values by ML model for the *i*-th data instance, $\bar{E}$ is the mean of the actual *ET* values, and n is the total number of *ET* forecasting made by ML model.
(3)$$R^{2}=1-\frac{\sum_{{}^{i=1}}^{n}{\left(E_{i}-\hat{E}\right)}^{2_{}}}{\sum_{i=1}^{n}{\left(E_{i}-\bar{E}\right)}^{2_{}}}$$

The Pearson correlation coefficient (r) is used to observe the similarity between the actual *ET* values in the data set and the forecasted *ET* values by the ML model. r is expressed by Equation (Equation 4), where X${}_{i}$ is set of meteorological conditions used as input to the ML model to forecast the *ET* for the *i*-th data instance, E${}_{i}$ is the actual *ET* value in the data set at *i*-th data instance, $\bar{x}$ is the mean of meteorological conditions, $\bar{E}$ is the mean of actual *ET* values, and n is the total number of *ET* forecasting made with the ML model.
(4)$$r=\frac{\sum_{i=1}^{n}\left(x_{i}-\bar{x}\right)\left(E_{i}-\bar{E}\right)}{\sqrt{\sum_{i=1}^{n}{\left(x_{i}-\bar{x}\right)}^{2}\cdot \sum_{i=1}^{n}{\left(E_{i}-\bar{E}\right)}^{2}}}$$

RMSE is used to observe the difference in forecasted *ET* value by the ML model from the actual *ET* value in the data set. RMSE is expressed by Equation (Equation 5), where E${}_{i}$ is the actual *ET* values in the data set at the *i*-th data instance, $\hat{E_{i}}$ represents the forecasted *ET* values for the *i*-th data instance, and n is the total number of *ET* forecasting with the ML model.
(5)$$RMSE=\sqrt{\frac{1}{n}\sum_{i=1}^{n}{\left(E_{i}-\hat{E}\right)}^{2}}$$

MSE is the mean of the squared difference between the forecasted *ET* values with the ML model and the actual *ET* values in the data set. MSE is expressed by Equation (Equation 6), where E${}_{i}$ is the actual *ET* values in the data set at the *i*-th data instance, $\hat{E_{i}}$ represents the forecasted *ET* values for the *i*-th data instance and n is the total number of *ET* forecasts made with the ML model.
(6)$$MSE=\frac{1}{n}\sum_{i=1}^{n}{\left(E_{i}-\hat{E_{i}}\right)}^{2}$$

The performance metrics of different ML models used for *ET* forecasting are given in Table 1. The ML algorithms considered for *ET* forecasting are off-the-shelf LSTM and ensemble LSTM with a bagged and boosted approach.

The results indicate that the ensemble LSTM algorithms performed well in forecasting *ET*, with decreasing error compared to the LSTM algorithms. The ensemble LSTM models also outperformed the off-the-shelf LSTM in achieving the lowest MAE and RMSE values of *ET* forecasting. The coefficient of determination (R${}^{2}$) and Pearson correlation coefficient (r) values were consistently high for the ensemble LSTM algorithms, indicating strong relationships between predicted and observed *ET* values. In the case of the ensemble LSTM approach, the bagged outperformed the boosted approach with high coefficient of determination (R${}^{2}$) and Pearson correlation coefficient (r) values and low RMSE, and MSE compared to the boosted approach. Overall, the performance of the bagged LSTM model is better in *ET* forecasting using limited meteorological conditions compared to the boosted and off-the-shelf LSTM approaches.

For comparison purposes, the *ET* forecasting with the ML model and actual *ET* values in the data set calculated using the PM method are shown in Figure 14, where it is evident that *ET* forecasting with the ensemble LSTM model from the test data set is more similar to actual *ET* values calculated by PM method compared to the *ET* forecasting with the off-the-shelf LSTM model. As concerns the ensemble LSTM models, the *ET* forecastings with bagged LSTM are more similar to the *ET* by PM method compared to the boosted LSTM.

Furthermore, Figure 15 illustrates the difference in *ET* forecasting with different models compared to the actual *ET* values determined by the PM method in the test data set. The absolute difference in actual *ET* values calculated by the PM method (*E${}_{i}$*) against the forecasted *ET* ($\hat{E_{i}}$) for each data instance ’i’ in the data set by each ML model is determined by Equation (Equation 7) and plotted in Figure 15.
(7)$$Difference=Absolute(E_{i}-\hat{E_{i}})$$

It is observed that the absolute differences in *ET* forecasting with the actual *ET* values with the PM method are low in the case of ensemble LSTM compared to the *ET* forecasting with the off-the-shelf LSTM model. In the case of the ensemble approaches, the differences in *ET* with the bagged LSTM with the *ET* by PM are fewer compared to the boosted approach.

## 4. Discussion

The performance of ensemble models in the forecasting of *ET* is attributed to their ability to handle nonlinear complex relationships between meteorological conditions and *ET*. The ensemble LSTM model is implemented with a bagged and boosted approach. The performance of the *ET* with different configurations of LSTM algorithms is observed in terms of RMSE and MSE in *ET* forecasting. The evaluation of all models on the test data set is made using 30% of the data set. The bagged LSTM ensemble model outperformed with high R${}^{2}$ of 0.94 and low RMSE and MSE of 0.42 and 0.53 compared to the boosted ensemble LSTM and off-the-shelf LSTM model.

The study also proposed LoRaWAN-enabled architecture for real-time meteorological data sensing and ensemble LSTM models with bagged and boosted LSTM to forecast *ET* with limited meteorological conditions. The LoRaWAN enabled the sensing of data from remote areas without a public network and the Internet. The real-time data enable accurate *ET* forecasting with fewer meteorological conditions. The meteorological conditions considered for the proposed solution are daily maximum temperature (*T*), mean humidity (*Hm*), and wind speed (Ws).

*ET* forecasting with limited meteorological conditions has significant value for water management practices. The proposed solution overcomes the difficulty associated with the implementation of *ET* for large meteorological conditions with high accuracy in *ET* forecasting. The proposed solution is limited in terms of exploiting stacked LSTM and other ensemble deep learning models. Experimentation with other meteorological conditions and the use of other deep learning models is recommended for future work.

## 5. Conclusions

The study proposed *ET* forecasting in Riyadh, Saudi Arabia, with limited meteorological conditions using LSTM and ensemble LSTM models. The meteorological data from the years 2001 to 2023 for Riyadh, Saudi Arabia, was used for the training and evaluation of ML model in *ET* forecasting. The bagged LSTM outperformed the boosted LSTM and off-the-shelf LSTM in *ET* forecasting with limited meteorological conditions in Riyadh, Saudi Arabia. The proposed solution has several applications in irrigation water management by dealing with the complexity associated with *ET* forecasting. The proposed approach has limitations in terms of the exploration of limited ML algorithms. The exploitation of more ML models and implementation in other parts of the world is recommended for future work.

## Figures and Tables

**Figure 1 sensors-23-07583-f001:**
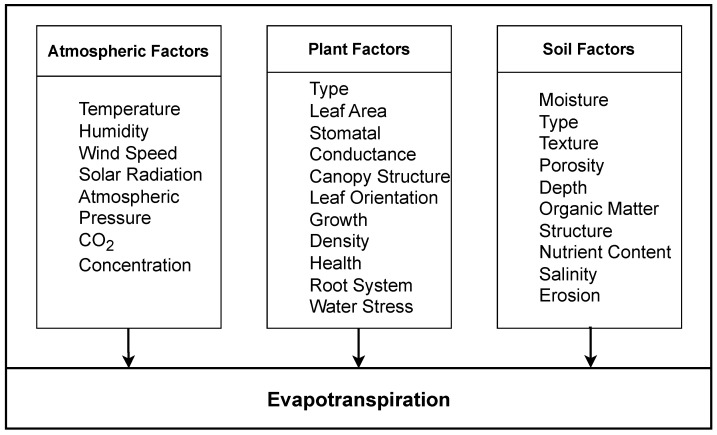
Factors affecting evapotranspiration (*ET*).

**Figure 2 sensors-23-07583-f002:**
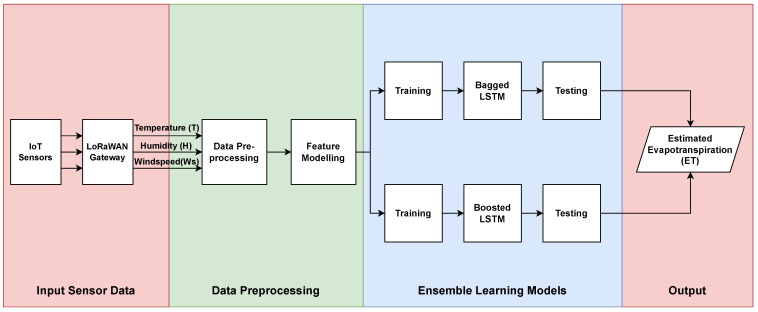
Flowchart of proposed *ET* forecasting.

**Figure 3 sensors-23-07583-f003:**
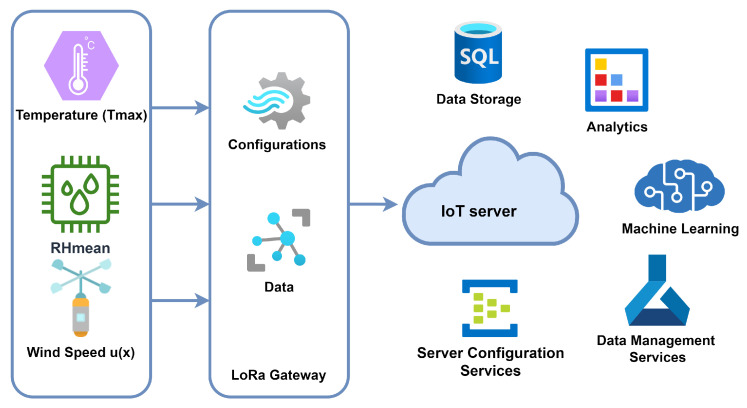
Proposed LoRaWAN-enabled IoT architecture.

**Figure 4 sensors-23-07583-f004:**
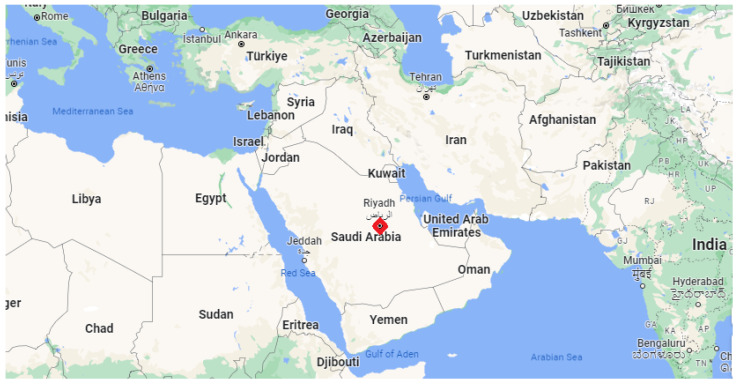
Location of Riyadh, Saudi Arabia on world map.

**Figure 5 sensors-23-07583-f005:**
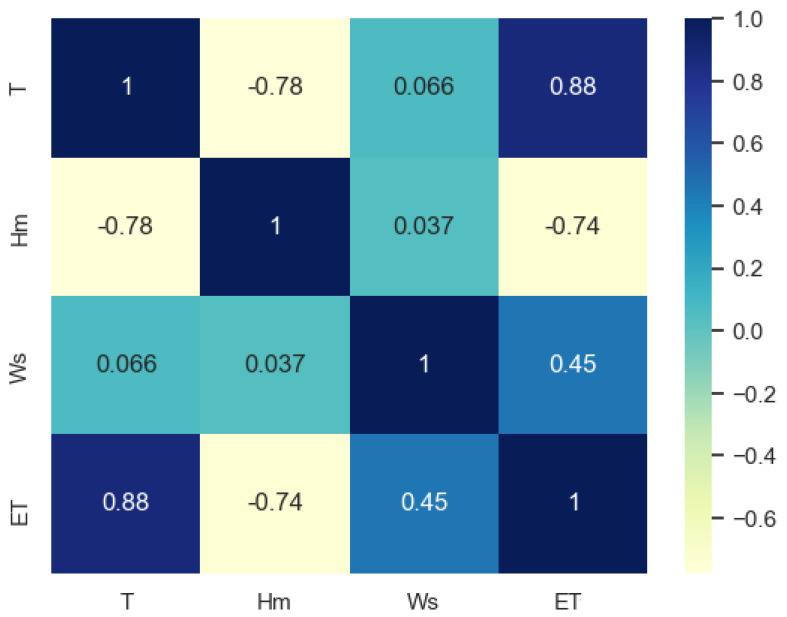
Correlation between climatic conditions and *ET*.

**Figure 6 sensors-23-07583-f006:**
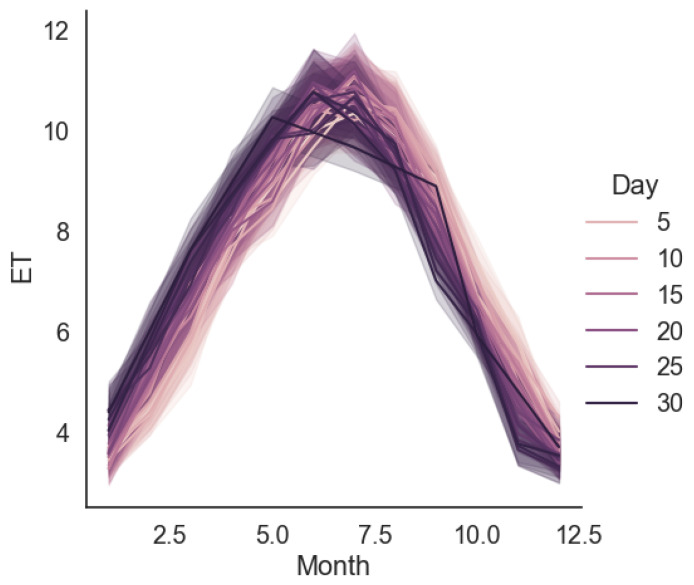
*ET* distributions in data set.

**Figure 7 sensors-23-07583-f007:**
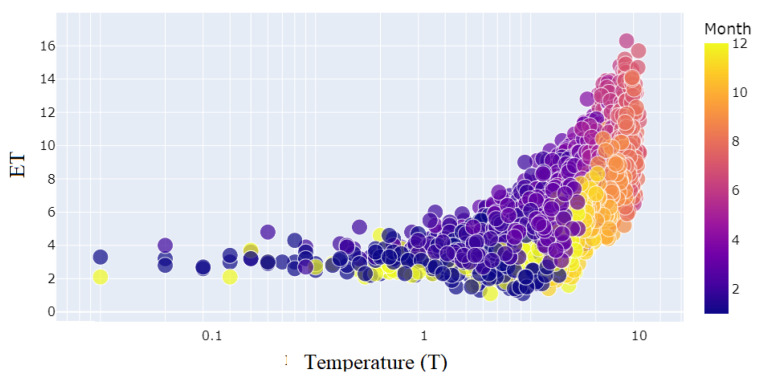
Distribution of daily maximum temperature (*T*) to months.

**Figure 8 sensors-23-07583-f008:**
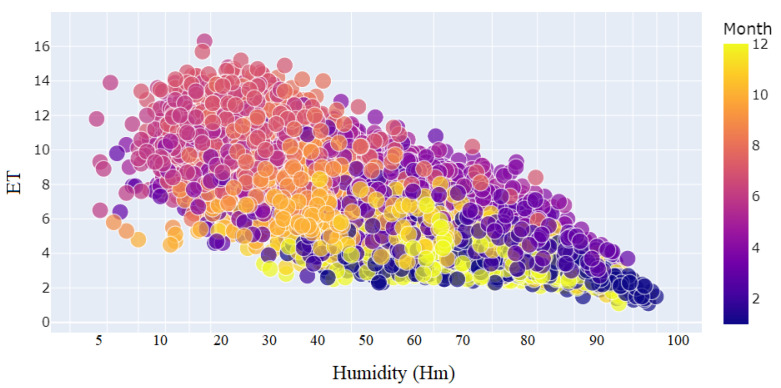
Distribution of daily mean humidity (*Hm*) to months.

**Figure 9 sensors-23-07583-f009:**
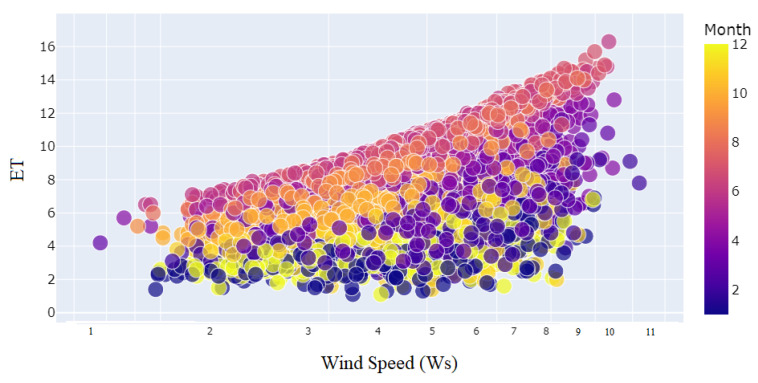
Distribution of daily maximum wind speed (*Ws*) to months.

**Figure 10 sensors-23-07583-f010:**
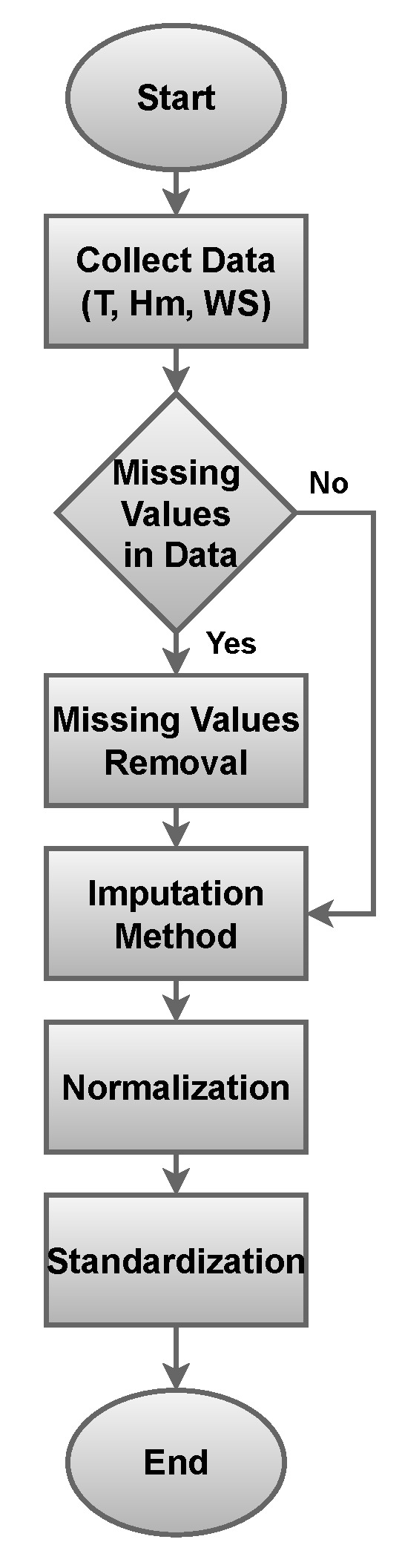
Preprocessing of data.

**Figure 11 sensors-23-07583-f011:**
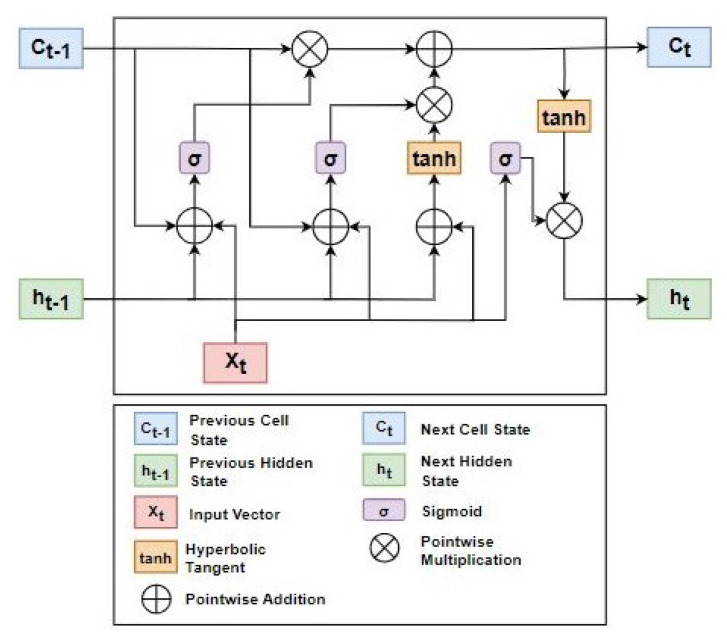
Architecture of LSTM.

**Figure 12 sensors-23-07583-f012:**
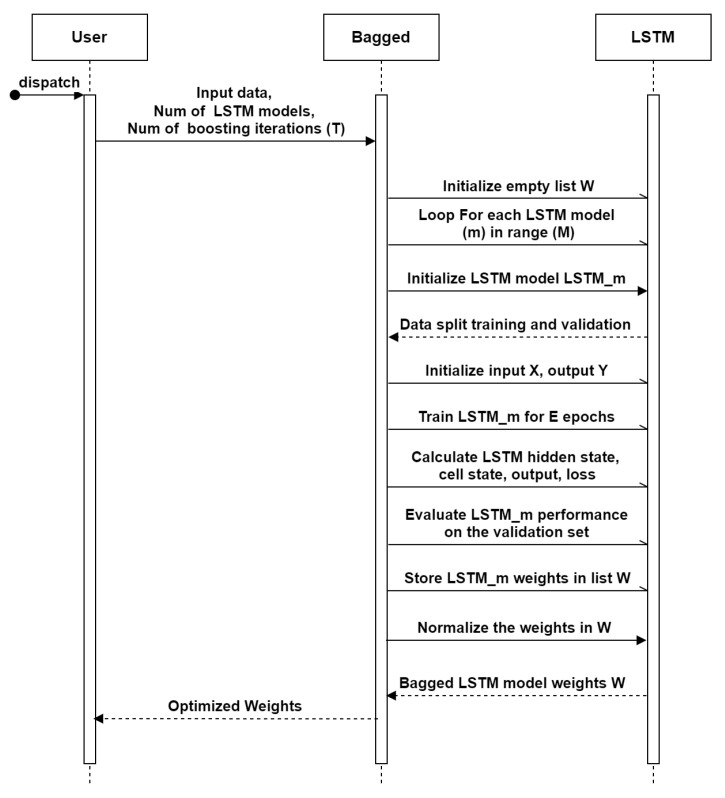
Sequence diagram of bagged LSTM.

**Figure 13 sensors-23-07583-f013:**
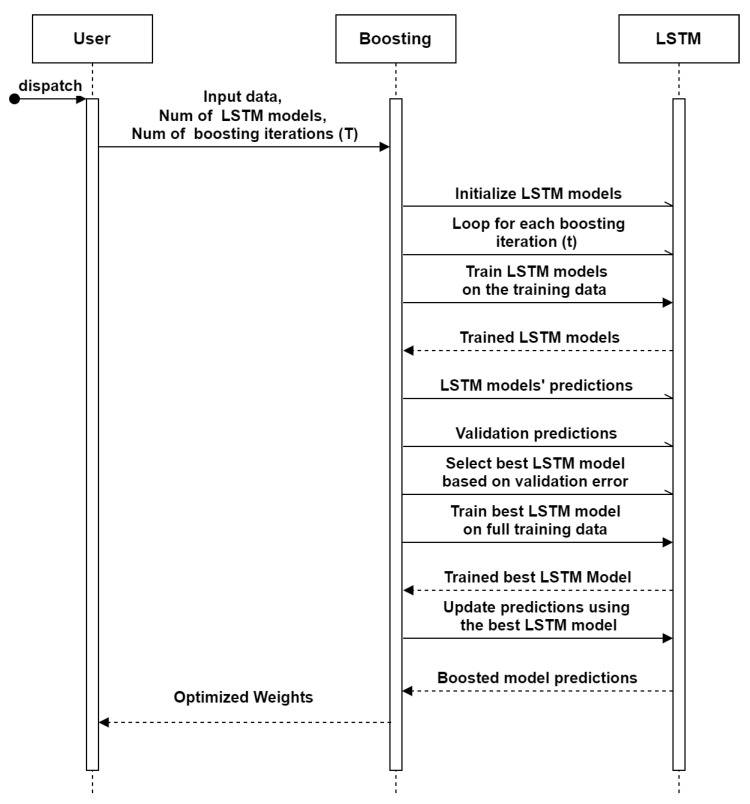
Sequence diagram of boosting LSTM.

**Figure 14 sensors-23-07583-f014:**
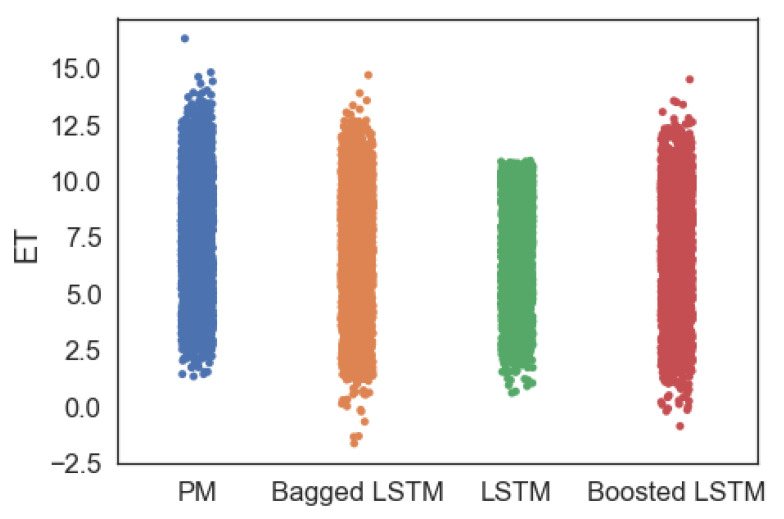
*ET* forecasted with different ML models along with actual *ET* values calculated by PM method in the test data set.

**Figure 15 sensors-23-07583-f015:**
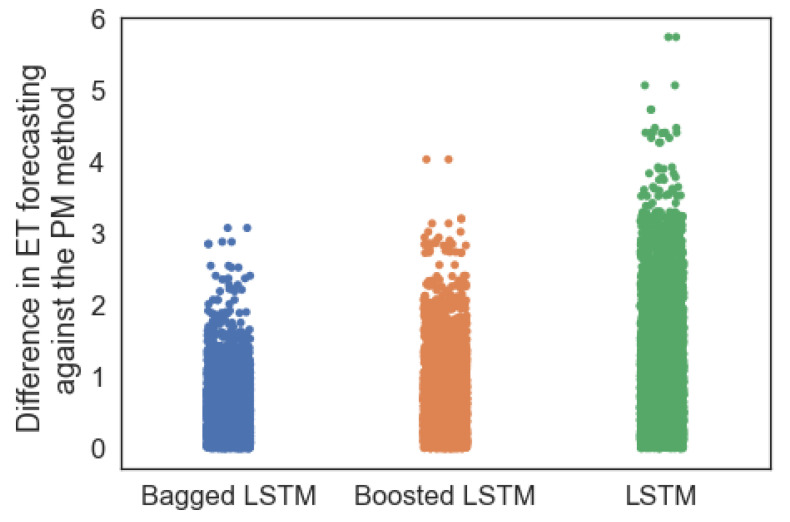
Difference in *ET* forecasted with different ML models compared to actual *ET* calculated using PM method.

**Table 1 sensors-23-07583-t001:** Performance Matrices of ML Models.

ML Model	R${}^{2}$	r	MSE	RMSE
Bagged LSTM	0.94	0.97	0.42	0.53
Boosted	0.91	0.95	0.63	0.63
LSTM	0.77	0.87	1.77	1.04

## Data Availability

Data will be available on request.

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
