# Peer review of "IoT and Ensemble Long-Short-Term-Memory-Based Evapotranspiration Forecasting for Riyadh"

_sensors, 2023, doi:10.3390/s23177583_

Round 1

Reviewer 1 Report

 The paper proposes a forecasting model with limited meteorological conditions to determine plant evapotranspiration, as a decision-making factor in irrigation systems. Three meteorological variables Tmax, RHmean, and u(x) are considered for ET forecasting, which were collected using IoT technology. LSTM, bagged and boosted ensembles LSTM ML approaches are implemented and evaluated to forecast the ET values from real-time meteorological data.

Related work is presented related to the specific problem addressed by the paper. Most approaches use ANN, SVM and other, but none using LSTM (or variations on LSTM) is presented, which I believe is very important to cover the related literature approaches. I suggest an extension of this section with emphasis on datasets descriptions, methods and materials and results obtained.

The paper does not explain the rationale for using LSTM based models with examples and references from the literature review.

Please explain Figure 5, as it is unclear the values and measurements used for each axis of each subfigure. Perhaps it could be easier to visualize all time series in one graphical representation.

Could u(x) be explained in more detail what is measures, is there a formula, what is it u(x) if no formula etc.

In section 2.3 data pre-processing methodology is presented. In terms of outlier removal, two approaches are mentioned – how have they been applied together, in which order. Please provide more details on how they were applied. Also, 3 imputation methods are enumerated, but there is no explanation on which was applied and how, what type of missing values etc. Are they all together applied on the data (?) or one by one in more experiments? It is very vague.

The algorithm for Bagged and Boosted LSTM is provided. How does this differ to other implementations of bagged and boosted LSTM? Please compare this with literature and emphasize only original contributions.

Existing references are appropriate and involve recent works, but incomplete in terms of LSTM.

Author contribution, funding, data and other required sections are not completed.

The topic and results obtained present great interest, but the paper requires a major revision to clarify the above mentioned points, in order to be valid for publication.

Author Response

Thank you for your valuabletime and suggestion to improve the quality of the work. Here is response to your suggestions.
Comment 1
Related work is presented related to the specific problem addressed by the paper. Most approaches use ANN, SVM and other, but none using LSTM (or variations on LSTM) is presented, which I believe is very important to cover the related literature approaches. I suggest an extension of this section with emphasis on datasets descriptions, methods and materials and results obtained.
Response : 
Agree, The desired section is updated with addition of new literature numbered 38,39,40,41,42 and enhancemnts in existing approaches discussed in the previous versions at Page 3 and 4 from line 100 to 141
Comments 2
The paper does not explain the rationale for using LSTM based models with examples and references from the literature review.
Response : The rational for using the LSTM is added at pasge  and line number 148 to 156 and on Page 11 at line number 255 to 258 
Comments 3
Please explain Figure 5, as it is unclear the values and measurements used for each axis of each subfigure. Perhaps it could be easier to visualize all time series in one graphical representation.
Response : Fig 5 is updated and repllaced with new figure in one grapphical format and additions of axis labels at Page 7
Comments 6
Could u(x) be explained in more detail what is measures, is there a formula, what is it u(x) if no formula etc.
Response : 
Agreed the submol used for Wind speed (ux) is changed to more appropriate sysmbol WS in whole manuscript
Comments 7
In section 2.3 data pre-processing methodology is presented. In terms of outlier removal, two approaches are mentioned – how have they been applied together, in which order. Please provide more details on how they were applied. Also, 3 imputation methods are enumerated, but there is no explanation on which was applied and how, what type of missing values etc. Are they all together applied on the data (?) or one by one in more experiments? It is very vague.
Response : 
Agreed, The section is improved with detailed discussion on page 9 and line 227 to 248
Comments 8
The algorithm for Bagged and Boosted LSTM is provided. How does this differ to other implementations of bagged and boosted LSTM? Please compare this with literature and emphasize only original contributions.
Response
Ok, a dsiciption of use of bagged and boosted and how they value to the contribution ofwork isadded at page No 11 and line number 256-258
Comments 9
Existing references are appropriate and involve recent works, but incomplete in terms of LSTM.
Response : Agreed, New refrences are added with refrences number ,39,40,41,42 
Comments 9
Author contribution, funding, data and other required sections are not completed.
Agreed, The required information added.
Comments 10
The topic and results obtained present great interest, but the paper requires a major revision to clarify the above mentioned points, in order to be valid for publication.
Response 
The paper is revised and improved by incroporating suggestions

Reviewer 2 Report

Review Comments

In this paper, an IoT system is used to measure meteorological data and combine it with machine learning algorithms to predict evapotranspiration, the research is more meaningful. However, there are a lot of deficiencies in the introductory part of the article, the results and discussion part is not rich enough, and there is a lack of comprehensive evaluation of the model prediction results, so it is recommended that the authors revise and organize the relevant parts. The specific comments are as follows:

1. there are problems with the format of references, and it is recommended to modify and organize them (e.g. font format of DOI links and years, etc.) to ensure the consistency and accuracy of the format.

2. the introduction section lists many factors that affect evapotranspiration, but does not explain why maximum temperature, average humidity, and wind speed were ultimately chosen as input factors, and it is recommended that the authors analyze this section to highlight the importance of these factors for evapotranspiration.

3. this paper does not analyze the existing empirical models used for evapotranspiration prediction. Empirical models also have a wide range of applications in evapotranspiration prediction, and it is recommended that the authors analyze the empirical models, including their strengths and limitations, and compare them with the machine learning model proposed in this paper.

4. In the article, many machine learning models are proposed and their related research results are analyzed and demonstrated, but the cited literature on related research is relatively small, and it only shows that machine learning has good performance in evapotranspiration prediction, and it does not analyze the superiority of the selected model LSTM in evapotranspiration prediction, It is suggested that the authors conduct a detailed analysis, including an explanation of why the LSTM model was chosen and its advantages over other machine learning models.

5. Please provide a more detailed analysis and discussion of the model prediction results, including comparisons between different models, error analysis, and credibility of the results.

6. The authors are requested to double-check whether the terms in the definitions of abbreviations are all used correctly in the text to ensure the accuracy and consistency of the abbreviations.

7. The text mentions the use of meteorological data to determine evapotranspiration through Penman's formula, but there is no detailed introduction of Penman's formula. Please add a detailed introduction of Penman's formula, including the content of the formula and the principle, etc., to ensure that the readers have a full understanding of this method.

8. The authors are requested to supplement Figure 10 by adding vertical coordinates and ensuring that the chart is clear and easy to read.

9. Please give the units of the coordinate axes of all the figures, and make sure that the information of the pictures is complete.

10. Terms that appear for the first time in the abstract should be used in their entirety, for example, replace "IoT" with "The Internet of Things", "LSTM" with "Long Short-Term Memory".

11. When discussing the advantages of ensembled machine learning algorithms, it is recommended that authors cite more relevant literature on the performance advantages of ensembled methods over original machine learning algorithms.

12. The authors are requested to discuss in detail the impact of existing research results on this study and the reasons that motivated the completion of this study, in order to highlight the innovation and contribution of this paper.

13. Please indicate the number of training and test datasets used in this paper so that the reader can understand the scale and reliability of this study.

14. The authors are requested to give a description of the software used to implement the machine learning algorithms and draw the figures in the paper.

15. In this paper, a comprehensive indicator evaluation model is missing in the discussion section of the model results, and the authors are requested to add a comprehensive indicator evaluation model in the discussion section of the model results.

16. It is recommended that some of the graphic content be modified to add color and graphic elements to make it more attractive and readable.

Author Response

Dear,
Thank you for your valuable time and suggestion to improve the quality of the work. Here is the
response to your suggestions.
1. there are problems with the format of references, and it is recommended to modify and
organize them (e.g. font format of DOI links and years, etc.) to ensure the consistency and
accuracy of the format.
Response
Agreed, The paper is revised and improved by incorporating suggestions to improve
reference style as per journal requirements
2. the introduction section lists many factors that affect evapotranspiration, but does not explain
why maximum temperature, average humidity, and wind speed were ultimately chosen as input
factors, and it is recommended that the authors analyze this section to highlight the importance
of these factors for evapotranspiration.
Response
Agreed, The rationale for the use of parameters is explained on page 8 at lines 223 to 226
with support of figure number 9.
3. this paper does not analyze the existing empirical models used for evapotranspiration
prediction. Empirical models also have a wide range of applications in evapotranspiration
prediction, and it is recommended that the authors analyze the empirical models, including their
strengths and limitations, and compare them with the machine learning model proposed in this
paper.
Response
Agreed, Empirical model penman Montieth is added on page 2 under Equation 1 with a
detailed description on page 2.
4. In the article, many machine learning models are proposed and their related research results
are analyzed and demonstrated, but the cited literature on related research is relatively small,
and it only shows that machine learning has good performance in evapotranspiration prediction,
and it does not analyze the superiority of the selected model LSTM in evapotranspiration
prediction, It is suggested that the authors conduct a detailed analysis, including an explanation
of why the LSTM model was chosen and its advantages over other machine learning models.
Response
Agreed the rationale to use LSTM is added n page 4 at line number 148 to 157
5. Please provide a more detailed analysis and discussion of the model prediction results,
including comparisons between different models, error analysis, and credibility of the results.
Response
Agreed, all the sections improved including discussion and results as per your
suggestion
6. The authors are requested to double-check whether the terms in the definitions of
abbreviations are all used correctly in the text to ensure the accuracy and consistency of the
abbreviations.
Response
Agreed, The terms, and abbreviations are checked
7. The text mentions the use of meteorological data to determine evapotranspiration through
Penman's formula, but there is no detailed introduction of Penman's formula. Please add a
detailed introduction of Penman's formula, including the content of the formula and the principle,
etc., to ensure that the readers have a full understanding of this method.
Response
Agreed, Empirical model penman Montieth is added on page 2 under Equation 1 with a
detailed description on page 2.
8. The authors are requested to supplement Figure 10 by adding vertical coordinates and
ensuring that the chart is clear and easy to read.
Response
Agreed, Figure 10 is improved with a label which is now Figure 14
9. Please give the units of the coordinate axes of all the figures, and make sure that the
information of the pictures is complete.
Response
Agreed, All the figures are updated
10. Terms that appear for the first time in the abstract should be used in their entirety, for
example, replace "IoT" with "The Internet of Things", "LSTM" with "Long Short-Term Memory".
Response
Agreed, the abstract is updated according to the suggestion
11. When discussing the advantages of ensembled machine learning algorithms, it is
recommended that authors cite more relevant literature on the performance advantages of
ensembled methods over original machine learning algorithms.
Response
Agreed, the required information is added on page 11 at line number 255 to 258
12. The authors are requested to discuss in detail the impact of existing research results on this
study and the reasons that motivated the completion of this study, in order to highlight the
innovation and contribution of this paper.
Response
Agreed, the discussion section is improved
13. Please indicate the number of training and test datasets used in this paper so that the
reader can understand the scale and reliability of this study.
Response
The paper is revised and improved by incorporating suggestions
14. The authors are requested to give a description of the software used to implement the
machine learning algorithms and draw the figures in the paper.
Response
Agreed, the manuscript is updated on page 10 lines 249 to 254
15. In this paper, a comprehensive indicator evaluation model is missing in the discussion
section of the model results, and the authors are requested to add a comprehensive indicator
evaluation model in the discussion section of the model results.
Response
Agreed, added on Pages 12 and 13 at lines 276 to 294
16. It is recommended that some of the graphic content be modified to add color and graphic
elements to make it more attractive and readable.
Response
Agreed, all the graph and pictures are updated attractive colors
Thank you again for your valuable suggestions.
Regards
Author

Reviewer 3 Report

Only a few minor corrections and recommendations:

Line 29 has a character ' that shouldn't be there.

Line 110 should be changed to "IoT-based."

Line 123 should change "relations" to "relationship."

Line 133 has a character ' that shouldn't be there.

Figure 4, Figure 5, Figure 6, Figure 7, Figure 8, Figure 9, and Figure 10 should be improved in vector format or similar.

While Algorithm 1 and Algorithm 2 are not bad, it would have been interesting to see a sequence diagram here.

The English needs improvement; there aren't many errors, but they are noticeable.

Author Response

Dear,
Thank you for your valuable time and suggestion to improve the quality of the work. Here is the
response to your suggestions.
Response Line 29 has a character ' that shouldn't be there.
Response
Agreed, removed
Line 110 should be changed to "IoT-based."
Agreed, updated as per suggestions
Line 123 should change "relations" to "relationship."
Agreed, updated as per suggestions
Line 133 has a character ' that shouldn't be there.
Agreed, updated as per suggestions
Figure 4, Figure 5, Figure 6, Figure 7, Figure 8, Figure 9, and Figure 10 should be improved in
vector format or similar.
Thank you for your valuabletime and suggestion to improve the quality of the work. Here is
response to your suggestions.
Response Line 29 has a character ' that shouldn't be there.
Response
Agreed, removed
Line 110 should be changed to "IoT-based."
Agreed, updated as per suggestions
Line 123 should change "relations" to "relationship."
Agreed, updated as per suggestions
Line 133 has a character ' that shouldn't be there.
Agreed, updated as per suggestions
Figure 4, Figure 5, Figure 6, Figure 7, Figure 8, Figure 9, and Figure 10 should be improved in
vector format or similar.
Agreed, updated as per suggestions with the update in all figures
While Algorithm 1 and Algorithm 2 are not bad, it would have been interesting to see a
sequence diagram here.
Agreed, updated as per suggestions with the addition of sequence diagram on pages 12
and 1 3 under figure number 12 and 13
Thank you again for your valuable suggestions.
Regards
Authors

Reviewer 4 Report

Minor revision

This paper mainly introduces the use of Internet of Things technology to collect meteorological data and combined with ML machine learning method to predict meteorological ET to make decisions. In short, this study is interesting and provides valuable results, but the current literature has several shortcomings. It must be strengthened to obtain document results of equal value to publications.

(1) The document contains a total of 34 employed references, of which 28 are publications produced in the last 5 years (82.4%), 2 in the last 5-10 years (5.9%) and 4 than 10 years old (11.8%) , implying a total percentage of 95% recent references.Generally speaking, the number of paper references is insufficient .

(2) The content of the introduction is too redundant, it is suggested that the author could be more concise, or be able to present some of the content in a later chapter.

(3) More state-of-art engineering applications may be discussed in the introduction. For machine learning prediction, please refer to An experimental investigation and machine learning-based prediction for seismic performance of steel tubular column filled with recycled aggregate concrete; Reviews on Advanced Materials Science; Prediction of thermo-mechanical properties of rubber-modified recycled aggregate concrete; Construction and Building Materials.

(4) It is suggested in Section 2.3 that the author can adopt the flow chart to make the data preprocessing more clear.

(5) In section 2.4, it is suggested that the author should describe the LSTM more carefully and use the form of image structure to describe the learning strategy.

Author Response

Dear,
Thank you for your valuable time and suggestion to improve the quality of the work. Here is the
response to your suggestions.

Round 2

Reviewer 2 Report

1. There is a problem with the format of the references. The author bolded most years of the paper, but there are still some years with different formats, so I hope the author can rearrange the format of the references.

2. There are still some abbreviations in the definition of abbreviations given in this paper that do not appear in the paper, please re-organize them.

Author Response

Dear,

Thank you for your valuable time and suggestion to improve the quality of the work. Here is the response to your suggestions.

Comment 1

  1. There is a problem with the format of the references. The author bolded most years of the paper, but there are still some years with different formats, so I hope the author can rearrange the format of the references.
    Response: Agreed, the references style is updated

    2. There are still some abbreviations in the definition of abbreviations given in this paper that do not appear in the paper, please re-organize them.

Response:

The abbreviations list is updated with the removal of some abbreviations and addition that are part of the manuscript.

Apart from these changes the grammar, and spelling is improved by proofreading.

Thank you

Authors
